# Challenges of Sharing REDD+ Benefits in the Amazon Region

**Raissa Guerra * and Paulo Moutinho**

Amazon Environmental Research Institute (IPAM), Belem 66093-672, Brazil; moutinho@ipam.org.br
* Correspondence: raissa.guerra@ipam.org.br

**Abstract:** The success of jurisdictional reducing emissions from deforestation and forest degradation (REDD+) initiatives is entirely dependent on how the REDD+ benefits are distributed among the stakeholders seeking to preserve the native vegetation and is considered one of the main challenges of REDD+. Among the existing benefit-sharing options, the adoption of the stock-and-flow approach to share REDD+ benefits has afforded fair jurisdictional systems in the states of Acre and Mato Grosso in the Brazilian Amazon. This innovative perspective is also the dividing line between inequitable and socially balanced jurisdictional REDD+ initiatives. However, these jurisdictions present challenges to fully implementing a robust benefit-sharing mechanism including the stock-and-flow approach and guaranteeing that resources will be accessible to the relevant beneficiaries. To better understand these challenges, we applied the Options Assessment Framework (OAF), a methodology proposed by the World Bank to evaluate the capacity to implement an effective benefit-sharing mechanism for REDD+, in Acre and Mato Grosso. The results indicated that these jurisdictions need to strengthen their conditions to guarantee the multi-faceted functionality of this mechanism and determine what aspects need more attention and where resources should be invested. Additionally, the results indicated that an equitable benefit-sharing mechanism is, by far, the main challenge faced by jurisdictions. Despite being a more evolved state in its REDD+ policies, Acre still needs to improve its institutional capacities, particularly in its local civil society organizations, to help communities access these benefits. The state of Mato Grosso, on the other hand, is still engaging in its REDD+ initiative and needs its institutional capacities to further mature to better organize its monitoring mechanisms and governance.

**Keywords:** REDD+; benefit-sharing mechanisms; stock-and-flow approach

## 1. Introduction

Reducing emissions from deforestation and forest degradation (REDD+) is a very promising method to handle deforestation-based greenhouse gas (GHG) emissions in developing countries [1,2]. It aims to offer results-based financial incentives to countries making efforts to reduce the emissions from deforestation and forest degradation in their territories [3]. Since 2006, this mechanism has been evolving. In 2009, the 15th Conference of the Parties (COP 15) in Copenhagen added the possibility to provide incentives for low-carbon sustainable development including forest conservation, sustainable management of forests, and increasing forest carbon stocks [4].Currently, the UNFCCC (United National Framework Convention on Climate Change) has adopted REDD+, and there have been many pilot initiatives around the world [5], yielding several discussions on the possible sources of funding to support those initiatives [5]. However, relatively little has been said about effective ways to share the benefits from REDD+ [6], which is the most important component to ensure the full operation of jurisdictional (national or subnational) initiatives in REDD+ [6–8]. A poorly designed and



implemented benefit-sharing mechanism has the associated risks of implementing ineffective, unequal, and/or inefficient REDD+ policies, as funds may not reach the targeted stakeholders [9].

The distribution of REDD+ benefits in Brazil follows specific rules, guidelines, and criteria defined by the National Commission for REDD+ (CONAREDD+ in Portuguese) and based on the recommendations from the Thematic Advisory Board. One of the main rules is not allowing performance-based payments to offset the country's mitigation commitments [10]. The REDD+ experiences in Brazil involve implementing performance-based mechanisms in which the benefits are distributed on the condition that the jurisdictions that receive the resources achieve a predefined standard of performance against a baseline.

As a condition for obtaining REDD+ payments, countries are required to develop a safeguard information system (SISREDD+) as a way to provide data on the implementation of REDD+ activities and monitor the respect for the relevant safeguards (Resolution 15, CONAREDD+ 2018). In Brazil, the REDD+ National Commission (CONAREDD+) is the entity responsible for developing the SISREDD. So far, the CONAREDD+ has developed and improved the guidelines for REDD+ safeguards [10] but not well enough to meet the expectations of the different groups of related stakeholders and social movements. There is still resistance in accepting REDD+ by social movements and organized civil society due to the risks of not having an equitable social share of benefits and other concerns related to land tenure, land rights, and access to natural and forest resources [11,12]. In many tropical countries, these rights are rarely well defined and can make indigenous and traditional populations excluded from the reward system [11]. Part of this resistance can be overcome if the REDD+ benefit-sharing mechanism were able to promote equitable rewards between those protecting and promoting the conservation of forests and those making efforts to reduce emissions from deforestation, allowing social groups to collectively benefit. The stock-and-flow approach (SF) is advantageous because it allows one to compensate for these two types of behaviors [2].

There is little research on understanding if benefit-sharing mechanisms in Brazil are taking place in an effective, efficient, and fair manner and on how jurisdictions are managing to structure themselves to implement those mechanisms. This study sets itself apart by taking a look at all aspects needed by jurisdictions to design robust benefit-sharing mechanisms and by listening to different sectors involved in REDD+. Considering the states of Acre and Mato Grosso are well advanced in their REDD+ jurisdictional systems, the main goal of this paper is to assess the challenges these states face in implementing effective, efficient, and equitable benefit-sharing mechanisms The results presented here are based on a report submitted to the World Bank, with support from the World Bank Program on Forests (PROFOR). This methodology has the advantage of shedding light on the main bottlenecks faced by jurisdictions when sharing benefits from REDD+ through the fulfillment of a list of fundamental criteria (institutional capacities, legal framework, monitoring, and fund management).

The results of this analysis will serve both decision-makers and researchers interested in analyzing the reasons for REDD+ successes and will contribute to the large discussion on how to develop effective, efficient, and equitable benefit-sharing mechanisms.

The analysis showed that good investments of resources and capacity building processes are still needed for beneficiary communities to become organized and access their benefits. They also indicated that this type of assessment needs to be constantly carried out to monitor the effectiveness, efficiency, and equity of benefit-sharing systems.

We organized this paper as follows. The first section introduces the problem. The second section provides a literature review on REDD+ challenges for benefit-sharing. Section 3 presents a literature review on REDD+ benefit-sharing. The fourth section outlines the benefit-sharing mechanism (stock-and-flow approach) as a requirement for REDD+ initiatives and the approaches used to share benefits. Section 5 provides the methods used to assess the capacities of states to implement robust benefit-sharing mechanisms. Section 6 discusses the results achieved, and in Section 7, we provide the main challenges and recommendations for developing robust benefit-sharing in REDD+ jurisdictional programs.

## 2. REDD+ Main Challenges

The REDD+ mechanism has undergone a long process of evolution. In 2005 (at COP 11) in Montreal (Canada), REDD+ was initially focused only on the reduction and degradation of forests, but today, it also considers forest conservation [13,14], sustainable management, and the enhancement of forest stocks [13], in addition to removing carbon via forests [15]. After more than a decade, REDD+ has made significant progress in its design. However, there are still a number of challenges that must be overcome for REDD+ to work efficiently, including the commoditization of developing countries' forests [16], leakage, permanence and additionality issues [17], weak funding [18], weak forest management institutions [18,19], the technical conditions needed to measure forest degradation events [20], transparency, and governance [21]. In addition, from the perspective of proponents on the ground, the main challenges faced are land tenure rights and REDD+ credit ownership, followed by the economic disadvantages of engaging in REDD+ activities, political interests, carbon market creation [5], and the capacity to fairly share benefits [6,22,23]. The latter is one of the most crucial challenges because it contributes to engaging people to take the desired actions to meet the ultimate goals of REDD+ [24]. Therefore, it is crucial that the benefit-sharing mechanism to be developed enables the recognition of local populations in REDD+ strategy and promotes their engagement.

## 3. Defining Benefit-Sharing Mechanisms and Its Main Challenges

Benefit-sharing (BS) is the reward (monetary or non-monetary) for achieving REDD+ action outcomes [25–27] and is considered one of the most important elements of the REDD+ mechanism because it provides positive incentives to the continued achievement of the final emissions reduction objectives [26]. This reward can be direct, as a direct result of the achievement of emissions reduction activities [25], or indirect, for the improvement of the provision of environmental services or even the improvement of infrastructure and the equipment necessary for the operation of REDD+ [25].

The beneficiaries include indigenous communities, riverine populations, family farmers, and even institutions that should be strengthened [26]. A robust BS mechanism must be effective (i.e., it must generate results), efficient (low cost), and equitable (fair) [26]. The latter is considered one of the biggest bottlenecks of BS [14,28] and needs special attention. However, if achieved, it would legitimize the REDD+ policy [6,29–33].

The major challenges of sharing benefits from REDD+ are reaching forest communities, especially from a gender perspective [34], defining the criteria to choose beneficiaries [35], understanding how can they be distributed in an effective, efficient, and equitable manner (3Es) [8,25,36] considering fundamental aspects such as fighting corruption, weak law enforcement, limited resources, and the conflicting objectives of different governmental policies [33,35], monitoring the flow of resources [8], and, finally, designing the necessary institutional arrangements and political instruments to organize the flow of resources [8].

For equitable distribution, it is fundamental to guarantee the effective participation of the stakeholders [22,36] and ensure they can participate in a qualified way, which will require long-term training effort [26]. Finally, the institutions that will implement REDD+ and share its benefits also need to be strengthened, which means having a strong financial management mechanism, a defined legal framework, and credible monitoring conditions [6,26,31].

There are methodologies aimed at detecting and overcoming the challenges of sharing REDD+ benefits. An example is the tree knowledge (TK), developed by the Center for International Forestry Research (CIFOR), aimed to understand the complexity of benefit-sharing in several different countries [37,38]. TK provides a contextual analysis of the jurisdictions and describes the steps taken to design benefit-sharing mechanisms. However, TK requires a long time and large funding to be implemented. There are also academic publications that aim to analyze the structures needed to achieve the 3Es needed for robust BS mechanisms [8,14,22,25,35,36], but there is still a lack of on-the-ground case studies focused on the main capacities to draw on benefit-sharing mechanisms.

## 4. Ensuring Equitable Benefit-Sharing for REDD+: The Stock-And-Flow Approach

The UNFCCC decisions on REDD+, other international frameworks and standards, and the leading organizations involved in REDD+ offer general requirements and guidance on how benefit-sharing arrangements should be designed and implemented but do not impose a particular approach [39]. Actually, each country is likely to have unique circumstances, preferences, and needs that will inform their benefit-sharing arrangements [27,36].

A set of criteria should be evaluated to ensure the efficiency of the benefit-sharing mechanism, such as existing institutional capacities, the presence of a legal framework necessary to implement those mechanisms, the ability to manage funds, and the technical capabilities to monitor forest and financial resources [6].

There are many strategies to consider when choosing an approach to share the benefits from REDD+. Each plan comes with a set of pros and cons, which must be carefully evaluated. One of these approaches is based on carbon stocks and is called the stock-only approach. This approach can produce negative effects and does not respect the principle of additionality, in which emissions would probably not occur in the areas under question, thus creating a false need to implement a REDD+ project [40].

Another way to share REDD+ benefits is through the flow-only approach, which proportionately benefits players based on their contributions to reducing emissions from deforestation but cannot monitor the direct contributions made by each player. The flow approach can prompt conflicts over social equity, especially regarding rights to land tenure. A promising possibility is the stock-and-flow approach (SF), which, in its purest form, consists of distributing funding to different land tenure categories according to their balanced contribution to stocks and reducing deforestation.

The SF approach in jurisdiction-wide REDD+ systems offers a set of advantages: (i) it promotes more significant social equity and political sustainability because it benefits both those who deforested in the past but have reduced their deforestations and those who have kept the forest standing; (ii) it permits a broader participation of society from the onset of discussions about the distribution; (iii) it enables one to create safeguards, which work better when benefits are shared equitably and can lead to a good level of respect for rights; and (iv) the SF approach offers flexibility in its operational tools and mechanisms and offers the possibility of improvements [40].

The SF was conceived in 2009 by a group of researchers from the Amazon Environmental Research Institute (IPAM) and The Woods Hole Research Center (WHRC) [41]; SF was considered a turning point in reaching political agreements on REDD+ due to its potential to fairly compensate all beneficiaries involved. As a consequence, the SF was adopted to regulate several REDD+ initiatives in Brazil, including the ICMS Verde in Pará state, the Acre´s State System of Incentives for Environmental Services (SISA), the SISREDD in Mato Grosso, and the national REDD+ strategy (ENREDD) [40].

The SF is oriented by two variables, deforestation data and forest stocks, which are measured by the different land category occupation information of the territory (e.g., indigenous groups, large private producers, rural settlers, and extractivists). This approach is, therefore, performed by calculating the forest carbon stock-and-flow reduction—that is, the carbon emissions for each land category in a given period—using a 50:50 ratio for the stock-and-flow values. The SF comes with structural challenges and distortions, such as providing benefits to stakeholders that detain large areas of stocks and do not contribute to flow reductions because they never promoted deforestation, which is the case for indigenous areas, or not fairly distributing land categories. If distortions are present, the proportions must be changed (40:60, for instance).

There are several examples of REDD+ initiatives based on benefit-sharing under the SF approach with specific adaptations. One of them is the National REDD+ Strategy in Brazil (ENREDD+), which mandates that the distribution of REDD+ benefits must be equitable for all the stakeholders engaged in reducing deforestation and degradation [42]. The CONAREDD+ Resolution n.6 of 6 July 2017 clearly defines that 40% of performance-based payments from emissions reductions must go to the federal government due to its efforts in maintaining native forests in protected areas (denominated as conservation units in Brazil) and indigenous lands, while 60% must go to the states of the Legal Amazon

(the states that are part of the Legal Amazon are Acre, Amapá, Amazonas, Maranhão, Mato Grosso, Pará, Rondônia, Roraima, and Tocantins) using the following criteria: (i) 30% of resources must go to states with occurrence of native forest (stock), and (ii) 30% must go to states with deforestation reductions (flow).

The SF approach is already being used in the REDD+ jurisdictional systems of Acre (AC) and Mato Grosso (MT). Acre has implemented the most advanced jurisdiction-wide REDD+ program in the world: the Environmental Services Incentives System (SISA in Portuguese) [43] and, in 2012, was awarded the REDD+ Early Movers Program (REM) from the Government of Germany through two contracts amounting to approximately USD 30 million, which were paid to the state as an incentive to avoid the emissions of 6.5 million tCO$_2$ for four years [40]. In 2017, Acre started the second phase of its REM strategy by signing its second contract (~USD 30 million). Since the 1980s, Acre has made many improvements to preserve its forests. These changes were mainly related to the design and implementation of innovative policies focusing on a low carbon economy, reduced deforestation, the protection of indigenous peoples, and economic development combined with social advances. In addition, Acre was a pioneer in implementing REDD+ jurisdictional systems. One of the main reasons for this forest preservation target is the paradigm that forests do not represent an obstacle to development. Instead, forests are seen as a chance to increase the income and well-being of local populations and the whole state. Therefore, many avant-garde initiatives were implemented by Acre, such as the concept of "forestship" (forest + citizenship), which promotes the inclusion of people from the forests in the state economy, and the creation of the Policy for Valuing Forest Environmental Assets that housed the SISA through the creation of State Law 2.308 [40].

The implementation of the SF in the Acre state was also innovative. To accelerate the program's distribution, the state took advantage of the preestablished structure of environmental initiatives and distributed benefits through existing programs, validated by the Validation and Monitoring State Commission (CEVA). Through this logic, called programmatic SF, instead of purely obeying the SF criteria, funds flow through preexisting programs [43]. The advantages include a decrease in implementation costs and an accelerated distribution process [2].

Following Acre, Mato Grosso signed its first contract (~USD 47 million) with the German and British governments for an REM strategy and also adopted the SF. A challenge that must be faced in the state is the great diversity of social actors distributed in various land categories that need to be included in the participative process of consultation. Moreover, Mato Grosso is currently one of the largest producers of commodities in the country. This activity has historically been linked directly or indirectly to the clearing of forests. Thus, we must find ways to reconcile economic growth related to agricultural activities with the conservation of the remaining forest stands [7].

The Climate Change Forum of the state collectively defined a rule for the distribution of its benefits. As a basic principle, the distribution of these REM benefits in Mato Grosso guarantees that 60% of the resources will be earmarked for actions involving the beneficiaries directly. The remaining direct benefits (40%) will be invested into the institutional strengthening of the state to guarantee the necessary conditions to implement and manage the activities foreseen in the SISREDD+ [44]. The resources will flow through four subprograms designed as part of the REM strategy: (1) The Family Agriculture, Peoples, and Traditional Communities in the Amazon Biome Program; (2) the Indigenous Peoples Program; (3) the Family Agriculture, Peoples, and Traditional Communities in the Cerrado and Pantanal Biomes Program; and (4) the Agriculture, Livestock, and Forest Management Program. Through this method, the final distribution will guarantee that 24.5% of the resources from REM go to family farmers located in the Amazon Biome, 13.2% to indigenous peoples, 12% to family farmers in the Cerrado and Pantanal Biomes, and 10.3% to medium and large farmers.

While these states are well advanced and aligned in the need to design robust REDD+ benefit-sharing systems, their implementation was followed by a number of challenges. Detecting such challenges is a fundamental part of working toward efficient benefit-sharing systems.

## 5. Methods

We evaluated the Acre and Mato Grosso states' capacities to implement effective benefit-sharing mechanisms for REDD+ using the Options Assessment Framework (OAF) [6], an evaluation methodology proposed by Pricewaterhouse Coopers (PwC) and adopted by the World Bank for assessing the capacities of a jurisdiction or a country through analysis of their institutional body, legal framework, fund management, and monitoring [6]. This methodology is useful because it is structured in a way to understand the gaps from the perspective of various stakeholders involved in the implementation of REDD+ and provides analysis all the necessary capacities for the proper functioning of a robust benefit-sharing mechanism. Furthermore, it is a relatively fast and low-cost methodology [6].

Using the OAF, we covered 42 key components organized into four thematic blocks of capacities (Table 1 and Appendix A): (1) institutional capacity (government, organized civil society, forest communities, and the private sector); (2) the legal framework for REDD+ and related themes; (3) capacity and experience in managing environmental funds; and (4) the capacity and experience in monitoring, especially in the case of forest cover and transparency in the use of financial resources through the wide dissemination of results and rendering of accounts.

**Table 1.** Building blocks of the Options Assessment Framework (OAF) to valuate benefit-sharing capacities.

| Four Thematic Blocks to Valuate Capacities | Components |
| --- | --- |
| Block 1. Institutional Capacities | Human resources, knowledge, experience levels, technical skills of personnel, physical presence in the field |
| Block 2. Legal Framework | National legislation and regulations relating to forest land ownership and tenure, allocation of forest rents, carbon ownership, national development plans, public access to information, and law enforcement. |
| Block 3. Fund Management | Fund management capacity and experience of organizations in the country, anticorruption mechanisms, andfund distribution networks |
| Block 4. Monitoring | Capacity and experience to monitor national or subnational programs, provision of frequent and publicly available monitoring reports about environmental spending programs, andstrong monitoring systems at the local and state levels (GIS tools). |

The OAF methodology was developed to identify how jurisdictions are, or not, prepared to implement efficient REDD+ benefit-sharing systems. The OAF uses a specific tool designed to help policy makers assess their country's readiness for the REDD+ benefit-sharing mechanism. This tool is part of a work supported by PROFOR for the Making Benefit-Sharing Arrangements Work for Forest-Dependent Communities: Insights for REDD+ Initiative [6], which is provided on an 'open source' basis as an excel file and contains the components of the analysis. It requires a three-researcher team to prepare the context reports, mobilize the actors, apply the questionnaires, systematize the answers, and analyze the results.

This methodology has the advantage of assessing the main challenges faced by countries to develop robust benefit-sharing mechanisms (institutional and regulatory frameworks and financial management and monitoring mechanisms) [6]. Methodological issues were raised during its implementation, such as the low relevance of some questions in the Brazilian context.

The questionnaire survey was applied to stakeholders directly involved with REDD+ at the national and subnational level with a deep technical knowledge of REDD+ and its mechanisms. Before the interviews, an introduction was presented to the stakeholders, explaining the goals of the assessment, the methods used, and the confidentiality of the information, ensuring that the names of respondents/institutions would be hidden, especially if some controversial subjects were raised.

The scores were arranged using three ordinal variables (0—no, 1—partially, and 2—yes), according to the methodological recommendations. For each question, a justification was required to support the score provided and improve the characterization of the components. Finally, the survey

form asked for concrete and feasible recommendations to strengthen the key components identified as problematic.

An important measure to make the assessment feasible was the segmentation of the key components according to each different interviewee profile (government agencies, civil society organizations, forest communities, and private sector), since the application of all these profiles would demand extended time, which most respondents could not afford. Moreover, some building blocks address very technical subjects and require specific knowledge; thus, consulting specialists was the best way to ensure the validity of the answers.

To complement the framework and capture issues that were not considered in the survey, at the end of the interview, an extra question was inserted: "Are there any other issues that were not mentioned but require attention to promote better functioning of the benefit-sharing mechanism?" This addition was vital to detect key issues not mentioned in the methodology.

In total, we conducted interviews with 44 stakeholders from different social segments related to REDD+ in the Acre and Mato Grosso states between May and July of 2018 (Table 2). In some cases, the interviews were conducted individually or with small groups. In other cases, we organized mini workshops with specialists on specific topics. All the meetings were recorded, transcribed, edited, and registered using a specific template. The majority of the meetings were conducted in person with specialists in each key component.

**Table 2.** Social segments interviewed in this survey and the number of interviews and interviewees.

| Organizations Interviewees | Acre | Mato Grosso | Total |
|---|---|---|---|
| Government | 6 | 3 | 9 |
| Organized Civil Society (OCS) | 6 | 4 | 10 |
| Private Sector | 2 | 2 | 4 |
| Community Forests * | 1 | 2 | 3 |
| International Cooperation | 1 | 1 | 2 |
| Total # of interviews ** | 16 | 12 | 28 |
| Total # of interviewees *** | 27 | 17 | 44 |

\* The community forests included representatives from indigenous quilombola groups, a traditional community; \*\* the mini workshops promoted with groups of different actors were counted only once; \*\*\* interviewees who accounted for more than one territory in the same conversation.

## 6. Results

The final scores were obtained from arithmetic averages, the contributions of experts, and the technical considerations of the team responsible for the evaluation. Finally, a subnational performance score was calculated in each thematic block, and a general score of all blocks was achieved. Figure 1 shows a graphical representation of the three different levels of performance achieved by the states of Mato Grosso and Acre. Scores of the capacities discriminated by key components in each thematic building block are provided in Appendix A.

Mato Grosso presented a greater proportion without the capacity to implement an adequate benefit-sharing system, while 38% of the capacities were partially fulfilled, and 21% were deficient or non-existent. On the other hand, Acre showed better performance: most of its main capacities had already been implemented and are currently in operation (74%), which indicates the maturity achieved over more than a decade of work towards a policy leading to REDD+. Otherwise, 12% of its capacities remained only partially fulfilled, and 14% were non-existent.

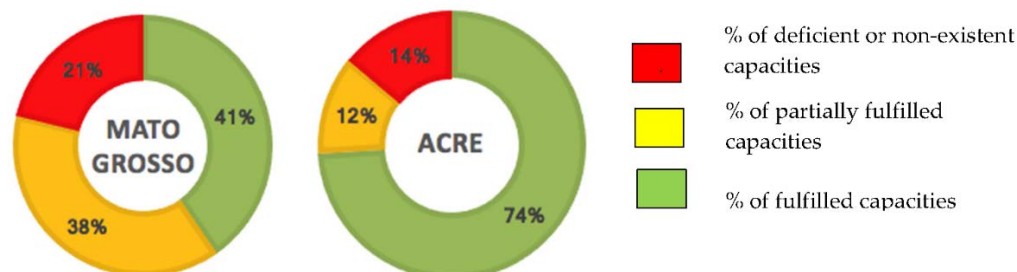

**Figure 1.** Comparison between the performance of Mato Grosso and Acre's capacities, based on the score attributed to the set of thematic blocks evaluated by the Options Assessment Framework (OAF) methodology (data from the OAF application).

A breakdown of the consolidated results is shown in Figure 2, which illustrates the extent of the capabilities of each jurisdiction per topic.

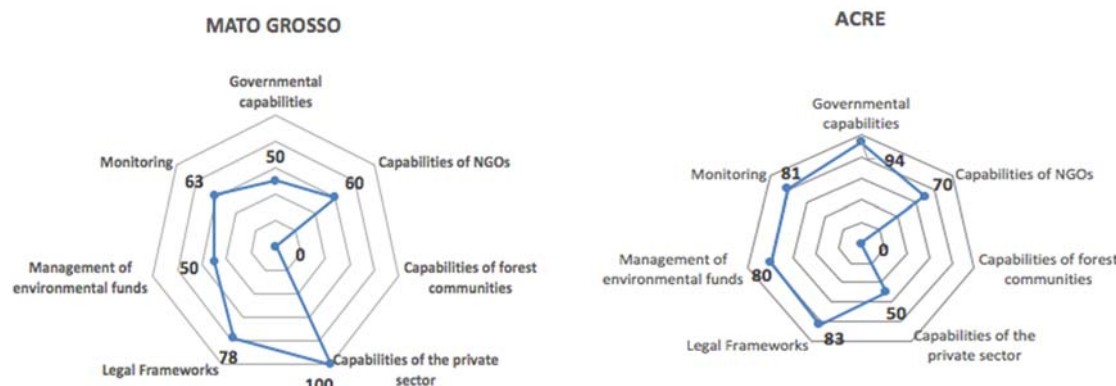

**Figure 2.** Performance of Mato Grosso and Acre's capacities, based on the scores attributed to the thematic blocks and the social players evaluated (data from the OAF application).

## 7. Discussions

This analysis shows, in a visual and simplified way, the level of each jurisdiction's capabilities per building block, thereby highlighting which capabilities need to be strengthened and which are already at an optimal level of development.

Acre showed the highest level of capacities developed to implement a REDD+ benefit-sharing system. The theme related to monitoring in the state of Acre was scored the best in the assessment framework, both in the thematic analysis and in the comparative analysis between jurisdictions. This reflects the efforts invested in enhancing the core of intelligence related to geographic information and the task force strategy among environmental institutions in data generation, storage, and processing.

Unlike Acre, the state of Mato Grosso has difficulties in generating spatial data and instead absorbs the national data, which may be too generic for a localized scale. Mato Grosso has a situation room that responds to extreme events but does not emphasize forest deforestation.

In Acre, SISA is an umbrella policy that includes seven programs, one of which deals with the incentives for the environmental services of carbon, which involves the REDD+ theme. The SISA law designates a Validation and Monitoring State Commission (CEVA), advised by a board of thematic councils (Environment, Forestry and Rural Development), to handle forest policy and decision-making involving the representatives of interest groups. The very active participation of all levels of stakeholders in Acre is one of its strengths and was inherited by the continuous political management in the last decade and the pioneering state of forest-based policies.

In contrast, the state of Mato Grosso had low scores in financial management and the administration of environmental funds and medium scores in the other thematic blocks. For the management of funds from the REM program, the German government has made a number of demands, including the

hiring of an external body to manage resources; this demand is being fulfilled by FUNBIO, which is a Brazilian private financial mechanism.

Traditionally, Mato Grosso was not a reference point for environmental protection. For many years, its deforestation rate was the largest in the Amazon and reached its peak in 2004, with an annual estimate of 12,000 km$^2$ deforested. In recent years, however, the state has been strengthening its actions to command, control, and structure its institutional and legal capacities to reverse this situation. The labor stability of the technical team involved in REDD+ activities can be an advantage for the state since it maintains the continuity of technical improvements even in the face of the political oscillations that periodically interrupt the management processes in Brazil and usually harm long-term environmental programs.

In both states, organized civil society has been working in parallel in solving the challenges of the government in empowering forest assets, although the scale of the Amazon territory and the recent financial crisis experienced by the country hamper full coverage of the demands presented. For both states, a great challenge is how to involve forest communities—indigenous or not—and introduce legitimacy and fairness of opportunity into the process. Amazonian peoples are marked by their cultural and linguistic diversity, diverse worldviews, and traditional practices with the forest, alongside their dispersion in a territory that is often inhospitable.

Thus, Acre has strong performance in its governmental capacities, likely due to the continuity of the REDD+ program in the state as a result of periods of political stability where the progress of the policies was absorbed and strengthened over longer periods through sequential government management. However, there are still some difficulties in achieving effective cooperation and political will between the federal and state governments on issues related to the conservation and management of forests.

Mato Grosso, on the other hand, has implemented REDD+ more recently; therefore, many of its aspects need to be strengthened to develop an efficient benefit-sharing mechanism. The most pressing points detected in the state at the time of the study were the lack of governmental structures to manage REDD+ initiatives and reach the local populations. Additionally, there was the need to improve the communication tools between the government and the local communities, followed by strengthening the internal capacities of the governmental agencies to implement the mechanism [7]. REDD+ is managed by a small team and must be disseminated to other secretaries. This weakness is currently being reinforced through the entry of financial resources by the REM program.

Forest communities are the segment with the lowest performance in terms of technical capacity to perform carbon monitoring, organize projects, and manage resources. In some cases, these groups do not even have the necessary documentation and access to banking institutions. In general, the members of this group need to enhance their representativeness and ability to influence themselves politically to ensure their rights and opportunities.

The private sector, another potential beneficiary from REDD+, performed well due to its capacities in the development of projects, monitoring, and the generation of comparative baselines between forest carbon, biodiversity, and socioeconomic factors but did not manifest interest in carbon markets. There is little interest of the private sector in engaging with REDD+ initiatives because forest conversion is a more profitable activity.

The legal framework of both states is highly advanced but does not provide a clear definition on the allocation of carbon rights related to land ownership. In 2010, Law n° 2308 introduced the State System of Incentives for Environmental Services (SISA), which includes Environmental Services Programs and State Ecosystem Products, including the Environmental Services Incentive Program (ISA Carbono), which contains the state REDD+ system. The Acre REDD+ system has guidelines that are convergent with other laws and programs, such as the National Plan of Climate Change (PNMC), the Ecological and Economic Zoning of the State of Acre—ZEE/AC (Law n° 1904/2007)—and the State Plan to Combat Deforestation in Acre (PPCD/AC), thereby establishing a good interoperational perspective.

In Mato Grosso, Law n° 9878/2013 introduced the REDD+ system. Despite having a well-developed legal framework, there are conflicts between the different legal instruments, which can compromise the understanding of REDD+. Moreover, in practice, the federal government possesses a single national calculation. For the monitoring block, high levels of capacities were observed in the area of natural resource monitoring technologies, but the transparency of the relevant information still needs to be improved.

In short, for these states to develop effective, efficient, and equitable BS mechanisms, it is necessary first and foremost to strengthen the relevant communities. In Mato Grosso, the institutional conditions need to be improved so that communication between the government and communities can be more effective. In the case of Acre, it is necessary to amplify the REDD+ activities in the communities and strengthen community capacity building to obtain greater engagement from local populations.

## 8. Conclusions

An effective, efficient, and equitable benefit-sharing mechanism is fundamental for improving the fairness and transparency of REDD+. The SF approach has high potential to achieve equity in the distribution process. The SF minimizes inequalities and promotes social equity, permits the participation of society in the overall discussion about benefits distribution, respects the relevant safeguards, and permits dialogue and flexibility in discussions to facilitate stakeholder agreements.

SF has been largely adopted in jurisdictional REDD+ systems in Brazil, especially in the Acre and Mato Grosso states. The experience in both states showed that it is possible to implement the SF and it has many advantages since it minimizes inequalities and promotes social equity, permits the participation of the society in the whole discussion about the benefits distribution, respects the safeguards, and permits dialogue and flexibility in the discussions to allow stakeholder agreements.

Through the OAF methodology, the capabilities of these two states to implement their benefit sharing systems were analyzed. Despite the large range of improvements, crucial challenges still remain. These jurisdictions suffer from institutional weaknesses and the absence of adequate governance to work in practice. Despite the country's political processes, relevant governance architecture has already been established and is immune to political party fluctuations that can anchor the processes initiated and implemented in the government guidelines.

There is an urgent need to invest in communities' capacities and expand their participation in REDD+ decisions. To do this, the institutional capacity of governments must be improved. There is also a need for goodwill from governments that are not always interested in REDD+ policies.

The uncertainty of the continuity of environmental policies is a recurrent problem in Brazil. Thus, it is essential that the necessary structures for the continuity of the construction of a REDD+ benefit-sharing mechanism are maintained. Although it is a complex construct whose time horizon exceeds governmental efforts, REDD+ remains a leading mechanism in national environmental service strategies and global policies for climate change.

**Author Contributions:** Conceptualization, R.G. and P.M.; methodology, R.G.; validation, R.G.; formal analysis, R.G. and P.M.; investigation, R.G.; resources, P.M.; data curation, R.G.; writing—original draft preparation, R.G.; writing—review and editing, R.G. and P.M.; visualization, R.G.; supervision, P.M.; project administration, R.G.; funding acquisition, P.M. All authors have read and agreed to the published version of the manuscript.

**Funding:** This research received funding from PROFOR/World Bank in the context of the activity "Improving social inclusion in the debate on the REDD+ benefit-sharing system under the context of the Brazilian REDD+ initiatives" through contract # 7,185,006.

**Acknowledgments:** The authors wish to thank Luciana Lopes for supporting the data collection. The authors gratefully acknowledge the opportunity to utilize data collected while working for the Program on Forests from the World Bank, which supported this research. The authors deeply appreciate all insights from the interviewees in the Acre and Mato Grosso states.

**Conflicts of Interest:** The authors declare no conflict of interest. The funders had no role in the collection, analyses, or interpretation of data; in the writing of the manuscript; or in the decision to publish the results.

## Appendix A. Performance of Mato Grosso and Acre Jurisdictions in Implementing a Performance-Based Benefit-Sharing Mechanism (Scores Range from 0 to 2 and Have Been Calibrated to Reflect Intermediate Situations)

| Color label: | | |
|---|---|---|
| | 🟥 | From inexistent to low capacities |
| | 🟨 | Medium capacities |
| | 🟩 | From high to full capacities |

| BUILDING BLOCK 1: INSTITUTIONAL CAPACITIES | MT | AC |
|---|---|---|
| 1. Proposed benefit-sharing mechanism implementation agencies (e.g., The Forestry Department and The Ministry of Environment) have sufficient technical forest management, community development, and technical REDD+ capacities to design and implement national-level benefit-sharing mechanism programs and associated activities. | 1 | 1.38 |
| 2. Existing and effective cooperation between national and subnational governments on sustainable forest management and conservation. | 1.00 | 1.08 |
| 3. Existing and effective coordination among all national agencies with mandates relevant to the proposed benefit-sharing mechanism (e.g., other sector agencies such as the Department of Agriculture). | 1.00 | 1.33 |
| 4. Proven capacity of the government to engage effectively with CSOs and the private sector for forest policy development and implementation at a centralized level. | 1.11 | 1.58 |
| 5. The physical presence and capacity of government offices with staff to engage and work effectively on forest policy and decision-making with community groups and the private sector. | 0.43 | 1.33 |
| 6. The intended benefit-sharing mechanism implementation agencies have the capability to store and process the financial, proprietary, and legal information needed to effectively administer a national scheme at a scale of millions of individuals and thousands of organizations. This includes tracking payment disbursals between different actors and beneficiaries in the benefit-sharing mechanism. | 1.67 | 1.67 |
| 7. Strong working relationship between the Department of Finance or Treasury and the benefit-sharing mechanism implementation agencies. Alignment of strategies and mandates among these bodies. | 1.00 | 1.33 |
| 8. Previous experience of the intended benefit-sharing mechanism implementation agency in communicating the purpose and function of national environmental programs and the eligibility criteria to the public in a timely and comprehensive manner. | 0.67 | 1.36 |
| 9. Presence and capacity of CSOs to support community groups and indigenous peoples in engaging in local forest-related planning, decision-making, and implementation. | 1.17 | 1.38 |
| 10. CSOs have a track record of working together with forest communities and helping those communities without formal land titles to access forest benefits. | 1.25 | 1.75 |
| 11. CSOs have the track record and capacity to assist forest communities with mapping, demonstrating, and registering their land rights. | 1.50 | 1.50 |
| 12. CSOs have sufficient forest management, community development, and technical knowledge and capacity to assist local communities to generate forest carbon, biodiversity, and socioeconomic baselines and to monitor against these baselines. | 0.33 | 0.43 |
| 13. CSOs have sufficient technical forest management, community development, and technical benefit-sharing mechanism knowledge and capacities to help the national benefit-sharing mechanism administrators distribute REDD+ benefits at the community level. | 1.83 | 1.36 |
| 14. Forest communities have sufficient technical forest management, conservation, and technical capacities to support, monitor, and report on local-level REDD+ programs and related activities in line with user-friendly guidance. | 0.50 | 0.55 |
| 15. The presence of a community of private-sector REDD+ project developers with the sufficient technical knowledge and capacity to generate forest carbon, biodiversity, and socioeconomic baselines and monitor against these baselines. | 1.43 | 0.78 |
| TOTAL SCORE OF THE BUILDING BLOCK | 15.66 | 18.82 |
| AVERAGE SCORE | 1.04 | 1.25 |
| RELATED SCORE | 0.52 | 0.63 |

| BUILDING BLOCK 2: LEGAL FRAMEWORK | MT | AC |
|---|---|---|
| 1. Recognition and enforcement of the customary or traditional forest rights of indigenous peoples, local communities, and traditional forest users in national legislation. | 1.00 | 2.00 |
| 2. Existence and enforcement of community forestry laws that give community groups management rights over forest land. | 1.33 | 1.50 |
| 3. The national forestry legislation clearly defines the allocation of forest rents to a forest rights holder dependent on the underlying land holding category (e.g., private land title, community land title, or concessionary land title). | 0.60 | 1.86 |
| 4. Clear and mutually supportive mandates given for all agencies involved with the proposed benefit-sharing mechanism. | 1.00 | |
| 5. The existence of effective coordination mechanisms to harmonize national development plans with the objectives of the proposed benefit-sharing mechanisms. | | |
| 6. The national legal framework fully supports public access to information, promotes debates related to forest policies, and imposes sanctions for failures to meet obligations to disclose information. | 1.40 | 1.50 |
| 7. Land rights legislation provides a clear definition of how forest carbon rights should be assigned according to land ownership. (xi) | 0.00 | 0.00 |
| 8. The existence and enforcement of a legal requirement in forest law to consult with, and gain consent from, communities for land-use decisions and benefit-sharing arrangements that affect the forest land for which they have customary or formal entitlement. | 1.00 | 1.40 |
| 9. National legislation defines the benefit-sharing arrangements between national, subnational, and local-level government institutions. (xii) | 2.00 | 2.00 |
| TOTAL SCORE OF THE BUILDING BLOCK | 8.33 | 10.26 |
| AVERAGE SCORE | **1.04** | **1.47** |
| RELATED SCORE | **0.52** | **0.73** |

| BUILDING BLOCK 3: FUND MANAGEMENT | MT | AC |
|---|---|---|
| 1. The presence of national government institutions or NGOs or private bodies with past experience in managing national environmental funds. | 0.67 | 1.50 |
| 2. The ability of community groups to open local bank accounts without onerous requirements (e.g., allowing community groups to open bank accounts without deposits) or other means of fund transfers. | 0.25 | 1.33 |
| 3. The presence of suitable fund management agencies with track records of managing forest revenue collection, budgeting, expenditures, accounting, redistribution, and audits. | 0.00 | 2.00 |
| 4. National codes of conduct and anticorruption measures are in place to safeguard against fund mismanagement. | 0.50 | 1.40 |
| 5. Track record of previous or existing environmental programs for disbursing funds to community groups or individuals at a national scale in a timely manner. | 0.50 | 0.33 |
| 6. The presence of third-party organizations with experience in providing financial and nonfinancial (e.g., governance) auditing of fund-management processes. | 2.00 | 2.00 |
| 7. The existence of effective and adequate standards against which the conduct of civil servants, political appointees, and community representatives can be held accountable, coupled with effective channels for reporting corruption and protecting whistleblowers. | 1.00 | 2.00 |
| 8. The presence of a national level government agency with experience in transferring monetary or nonmonetary benefits to beneficiaries linked to measurable and verifiable performance against predefined targets. (xiii) | 0.00 | 2.00 |
| 9. The existence of a government or a public/private organization with experience in managing environmental revolving funds. (xiv) | 0.00 | 0.50 |
| 10. The existence of a government or a public or private organization with experience in providing low-interest, long-horizon, risk-tolerant loans to community groups, members of the public, social enterprises, and the private sector. | 0.00 | 1.29 |
| TOTAL SCORE OF THE BUILDING BLOCK | 4.92 | 14.35 |
| AVERAGE SCORE | 0.49 | 1.44 |
| RELATED SCORE | 0.25 | 0.72 |

| BUILDING BLOCK 4: MONITORING | MT | AC |
|---|---|---|
| 1. The presence of organizations at a national level with a sufficient combination of experience monitoring forestry, social-orientation, and ecological conservation projects. | 1.75 | 1.60 |
| 2. Demonstrated ability of the government to provide frequent and publicly available monitoring evaluation reports on government environmental spending programs. | 0.57 | 1.11 |
| 3. Demonstrated ability to decentralize monitoring systems and transfer them to local or nongovernmental institutions to assist with benefit-sharing mechanisms and socioeconomic impact monitoring. | | |
| 4. The prior and effective use of third-party monitoring agencies within national government environmental programs. | 1.33 | 2.00 |
| 5. Proposed REDD+ benefit-sharing mechanism implementation agencies have experience with incorporating monitoring and evaluation data into forest management planning and using evaluation results to continually improve program implementation. | 1.33 | 1.60 |
| 6. The proposed benefit-sharing mechanism implementation agencies have experience with using GIS data to monitor changes in forest cover or have an existing partnership with a national-level organization with this capacity. (xvii) | 1.50 | 2.00 |
| 7. The proposed benefit-sharing mechanism agency has experience in using GIS data to monitor changes in forest cover and in using these data to calculate and monitor changes in bio-carbon stocks and abatement or has an existing partnership with a national-level organization with this capacity. | 1.00 | 1.00 |
| 8. The proposed benefit-sharing mechanism agency has experience in ground-truthing GIS data on forest-cover change or has an existing partnership with a national level organization that has this capacity. (xix) | 0.00 | 2.00 |
| TOTAL SCORE OF THE BUILDING BLOCK | 7.49 | 11.31 |
| AVERAGE SCORE | 1.07 | 1.62 |
| RELATED SCORE | 0.53 | 0.81 |

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
