# Peer review of "Challenges of Sharing REDD+ Benefits in the Amazon Region"

_forests, doi:10.3390/f11091012_

Round 1

Reviewer 1 Report

This paper presents an assessment of the capabilities of two regions of Brazil to implement REDD+ and share benefits fairly among stakeholders. The authors advocate the use of the Stock-Flow approach (SF) as the best system to share benefits and list a number of advantages over stock-only and flow-only approaches. However, they do not present enough detail for readers to understand exactly how this system works. They apply the Options Assessment Framework, which was developed by PwC and endorsed by the World Bank, so their methodology is defensible. The application to two states in the Amazon has the potential to provide useful lessons to practitioners and researches in other regions, however, this potential is not fulfilled in the paper.

The manuscript has two major weaknesses, (1) poor English writing, and (2) a lack of recommendations that may help improve not only the cases they study but also others around the world. As it stands the manuscript is simply an evaluation of two regions of Brazil and its contribution to the literature is not clear. What makes this paper worthy of publication in a peer reviewed journal when the World Bank report on which it is based could be already accessed by interested parties? There are very few references to the peer reviewed literature which reinforces the impression that the contribution is not clear.

Regarding point (1), the problem is easily fixed by having the paper carefully edited for proper English expression and grammar. As a reviewer I don’t see my role as helping the authors fix this, but I point out a few cases where this problem is obvious.

Line 31-32: ‘Conference of the Parts’ should be ‘Parties’.

Line 46: ‘Brazil’s experiences implement performance-based mechanisms’, should be ‘implementing’

Line 50: ‘more susceptible to social exclusion and excluded from the rewarding system’

Lines 75-76: The authors refer to section 5 twice and exclude section 4.

Lines 92-93: ‘does not respect the principle of additionality in which emissions would probably not occur in those areas creating a false necessity to implement a REDD+ project’. Poorly worded.

Lines 200-201: ‘The surveys were conducted in a primary research method (individual and collective 200 interviews)’. There is no need to mention that a primary research method was used, this is implied by the fact that surveys were conducted.

Line 203: ‘an introduction was exposed to stakeholders’ should be ‘presented’ rather than ‘exposed’.

Lines 234-235: ‘Some institutions requested to previous consult the questions’ should be ‘Some institutions requested to consult the questions prior to the interviews’

Line 236: ‘This was very assertive and provided agility’, it is not clear what ‘assertive’ means in this context.

Lines 240-241: ‘a subnational performance was calculated’, it should be ‘performance score’.

Line 265: ‘Acre was the well scored’, should be ‘Acre was scored the best’.

Line 277: ‘The capillarity of the discussions’, it is not clear what this means.

Line 285: ‘Mato Grosso is not a reference on environmental issues’, it is not clear what this means.

Line 300: ‘Acre had a great performance’, should be ‘strong’ rather than ‘great’.

These are just a few cases demonstrating that the paper needs a thorough review. There are many others in the manuscript.

Regarding point (2), the problem could be addressed by including a brief review of the challenges faced by REDD+ as published in leading journals, and demonstrating the contribution of their work to this knowledge. It seems that the authors have valuable expertise that could be better shared by revealing ways in which the weaknesses found in their case studies could help improve these programs, and putting these recommendations in the context of the literature. Another useful contribution could be to explain the challenges they faced in carrying out the OAF methodology. What sorts of resources, budget and time are required to complete a study of this type? Would this sort of evaluation be feasible only with support from the World Bank or similar agencies?

Finally, the manuscript has too many acronyms referring to Brazilian organizations and programs, it would be useful to have a table listing all these acronyms in one place.

Reviewer 2 Report

This article describes well the challenges of REDD+ benefit sharing in Amazon region.

However, the article lacks the followings. I suggest to add the following elements.

  1. Literature review on REDD+ benefit sharing. There are number of literature on this topic.
  2. Explanation of why your methodology is useful, by comparing with existing research approaches.
  3. Explanation of how your research contributes to the academic discussion related to REDD+.
  4. Explanation of how REDD+ benefit sharing is important in the current challenges of overall REDD+ framework. One of the current challenges of REDD+ is the lack of finance for REDD+. 

Round 2

Reviewer 2 Report

Although the authors added the explanations of challenges on REDD+ and benefit-sharing, the authors have not fully responded to my comments, which are important elements in writing academic articles.

Also the responds to the comments are not sufficient, and I read the lines you mentioned, but it was difficult to understand how you responded to the comments. 

This paper is a good report that explain and assess the projects you worked on, but this paper is not an academic paper yet. The paper lacks the description on research gaps, objectivity and novelty of the methods you used compared with existing studies, and contribution of this research to the academic research in this field.

1. Literature review on REDD+ benefit sharing.

Still lacks description on what kind of research has been conducted in this field, and what kind of research/research approaches are missing (research gaps)

2. Explanation of why your methodology is useful by comparing with existing research approaches.

There is no explanation of existing research approaches, so the method you used lacks the objectivity and novelty.

3. Explanation of how your research contributes the academic discussion related to REDD+. 

I cannot find the explanation on this. Still lacks the description of research gaps.

4. Explanation of how REDD+ benefit-sharing is important in the current challenges of all REDD+ framework.

Need to explain how the current challenges of REDD+ framework links to the issues in benefit-sharing. How weak funding, weak institutions, governance etc. link to the issues of benefit sharing?

5. The following goal of this paper needs to be revised to research questions. Needs to explain why this paper assesses the challenges of benefit-sharing mechanisms in a specific states.

The main goal of this paper is to assess the challenges in implementing effective, efficient, and equitable benefit-sharing mechanisms in the states of the Brazilian Acre and Mato Grosso.
